# Automated Protocol for Monitoring Droplets and Fomites on Surfaces

Federica Valeriani , Lory Marika Margarucci, Francesca Ubaldi, Gianluca Gianfranceschi and Vincenzo Romano Spica *

Laboratory of Epidemiology and Biotechnologies, Department of Movement, Human and Health Sciences, University of Rome 'Foro Italico', 00135 Rome, Italy; federica.valeriani@uniroma4.it (F.V.); lory.margarucci@tiscali.it (L.M.M.); f.ubaldi@studenti.uniroma4.it (F.U.); gianluca.gianfranceschi@uniroma4.it (G.G.)
* Correspondence: vincenzo.romanospica@uniroma4.it; Tel.: +39-0636733247

**Abstract:** During the COVID-19 pandemic, extensive efforts focused on developing a better understanding of indirect transmission routes, environmental monitoring of fomites, and suitable surveillance strategies, providing new perspectives to also face other communicable diseases. Rapid methods for monitoring environmental contamination are strongly needed to support risk assessment, epidemiological surveillance and prevent infections from spreading. We optimized and automatized a protocol based on fomite detection by qPCR, using a microbial-signature approach based on marker genes belonging to the microbiota of droplets or different biological fluids. The procedure was implemented by exploiting the available tools developed for SARS-CoV-2 tracing, such as flocked swab sampling, real-time PCR equipment and automatic extraction of nucleic acids. This approach allowed scaling up, simplifying, and speeding up the extraction step of environmental swabs, processing at least 48 samples within 45 min vs. 90 min for about 24 samples by manual protocols. A comparison of microflora data by Next-Generation Sequencing (NGS) strongly supports the effectiveness of this semiautomated extraction procedure, providing good quality DNA with comparable representation of species as shown by biodiversity indexes. Today, equipment for qPCR is widely available and relatively inexpensive; therefore this approach may represent a promising tool for hospital hygiene in surveilling fomites associated with SARS-CoV-2 or other pathogen's transmission.

**Keywords:** environmental surveillance; qPCR; fomites; 16S rDNA; automate nucleic acid extraction; microbiota



## 1. Introduction

The COVID-19 pandemic highlighted the need for prompt and simple tools to monitor the contamination of pathogens in the environment, providing new strategies for further advances in the field. Indeed, hygiene and surveillance tasks represented a worldwide shared background and an exemplificative case study to focus on risk assessment and prompt response by developing environmental monitoring strategies [1,2]. SARS-CoV-2, like other respiratory pathogens, is spread via respiratory droplets generated by infected persons while talking, coughing, and sneezing [2]. As for other microorganisms, the risk of transmission may also occur through other routes, such as indirect transmission via fomites on several surfaces, including furniture and fixtures [3–15]. Respiratory droplets (aerodynamic diameter ranging between 6 and 10 μm) and droplet nuclei or aerosols ($\leq$5 μm) can directly reach the mouth, nasopharyngeal cavity, or eyes of a susceptible person but can also land on surfaces, contaminating them [16–19]. Coronaviruses can survive on different matrices under different conditions, persisting from hours to days, especially in indoor environments [20–22]. Some of these ways may be undervalued, which can intensify spreading the virus. Knowledge about environmental contamination is important

during outbreaks and transition phases to enforce public health measures for both symptomatic and asymptomatic individuals [23]. Concerns about environmental contamination and the associated risks of indirect transmission can be raised in specific environments (e.g., hospitals or schools and public settings), where SARS-CoV-2 was detected even when sanitation measures were accurately performed [24,25]. Environmental surveillance, joined with other public health measures, such as contact tracing, clinical reports, and laboratory-based testing, represents a key strategy to assess indoor exposure to respiratory pathogens. Environmental monitoring of SARS-CoV-2 virus was also performed in different studies on other matrices, including air and wastewater [26–33]. From an epidemiological point of view, screening of environmental samples can also provide indirect evidence of the number of infected people shedding the virus in that surrounding area [34–36]. Monitoring the presence of pathogens in the environment has several advantages, such as providing an indirect framework for tracing outbreak diffusion and its evolution over time, identifying vehicles at risk for indirect transmission, and supporting surveillance and sanitation procedures. However, suitable indicators and methods are still needed to provide a comprehensive picture of the spreading of several respiratory viruses as well as other pathogens and antibiotic resistant bacteria [37,38]. Traditionally, the effectiveness of disinfection protocols is checked by using conventional microbiology methods, such as the observation of growth and colony morphology, biochemical characterization, or by other means, but all these strategies are aspecific, providing only a generic indication for the presence of organic contamination [2,37,38]. The use of environmental monitoring of fomites through qPCR was investigated to identify specific anthropic contaminations on hospital surfaces and allowed to detect the presence of SARS-CoV-2 in the surroundings of a hospitalized patient [2]. Environmental surveillance by qPCR may become a cost-effective complement to clinical testing, with several advantages in reducing critical issues associated with informed consent, anamnesis, and operational logistics that can constrain epidemiological programs during an outbreak or pandemic [33,39]. However, at present, clinical, and environmental tracing seems to follow distant and independent ways, involving completely different expertise and lacking guidelines for molecular or qPCR surveillance [40,41]. Regardless of future policies, the present main problem is related to the technical need to process simultaneously an elevated number of samples to address environmental surveillance in a specific site [2,39–41]. Therefore, simple wipe tests, effective storage procedures and higher throughput DNA/RNA extraction methods are strongly needed to overcome this bottleneck in sample processing. Several critical issues need to be considered when approaching environmental samples, including the presence of amplification inhibitors such as disinfectants or humic acids, and the possibility to access protocols, know-how, equipment and materials that are already available. The purpose of the present study is to test the possibility of processing a higher number of environmental swabs by qPCR, reducing the time-consuming extraction step by automatization. We focused on evaluating and scaling up a multiplex qPCR protocol for detecting droplets and fomites in the environment by a microbial signature approach.

## 2. Materials and Methods

### 2.1. Study Design

To define and test a fast and high-throughput protocol for detecting traces of droplets or biological fluids in environmental specimens, a set of mock samples containing different biological fluids was prepared and aliquoted for assessing reproducibility during intra- and inter-laboratory trials. We used the same flocked swabs used for clinical tests or contact tracing to consider ready readily available materials. Moreover, this sampling approach was previously tested and compared, showing high effectiveness not only in clinical and environmental surveys but also in forensics [42–44]. Collection medium-eNat® (COPAN Italia Spa, Brescia, Italy) were contaminated with biological fluids (nasopharyngeal, saliva, skin, feces collected from six independent healthy volunteers), as reported in Table 1. Reagents for processing the samples included: DNA extraction kit (QIAmp

DNA Mini Kit and DNeasy Blood & Tissue Kit, Qiagen, Hilden, Germany); Nucleic Acids Optimizer (NAO) baskets (COPAN Italia Spa, Brescia, Italy); and lysozyme solution and glass beads (Sigma Aldrich, St. Louis, MO, USA). Each operator received the protocol, the reagents from the same batches, and the set of aliquots to be tested. In two different laboratories, independent operators performed the reproducibility test following biosafety laboratory procedures (BSL2 level). Anthropic contamination was assessed by searching for biological fluids through the detection of their microbiota traces by qPCR, as previously described [2,42,45–50]. After evaluating the qPCR approach by different operators in two independent laboratories, a semiautomated protocol was tested on the same spiked sample aliquots (duplicate for each laboratory) and on environmental samples (duplicate for each type of surface) (Table 1). Surface sampling was carried out on different surfaces (handlebars of bicycles or exercise bikes, gymnastic rings, keyboard, microphones, napkins, headphones, table cutlery) after use and/or prolonged exposure to human presence (at least 4 h). Briefly, FLOQSswabs® (COPAN Italia Spa, Brescia, Italy) were wiped in two perpendicular directions, changing the faces of the flocked tip, and covering a 3–5 cm² surface area, as previously described [50–53]. FLOQSwabs® were immediately soaked into eNat®, (COPAN Italia Spa, Brescia, Italy), a molecular medium that inactivates microbial infectivity and preserves DNA and RNA nucleic acids for up to four weeks at room temperature or at 4 °C (and up to 6 months at −20 °C to −80 °C) and analyzed within 15 days in parallel using both the Manual protocol and by an automated instrument (Nextractor-48S system, Genolution, Seul, Republic of Korea).

**Table 1.** Data set of mock contaminated and environmental samples.

| Type of Biological Fluids | N | Description of Spiked Samples |
|---|---|---|
| Nasopharyngeal high concentration (A) | 2 | Nasopharyngeal mix |
| Nasopharyngeal low concentration (B) | 2 | Nasopharyngeal mix (Diluted 1-fold) |
| Saliva high concentration (A) | 2 | Saliva traces mix |
| Saliva low concentration (B) | 2 | Saliva traces mix (Diluted 1-fold) |
| Skin high concentration (A) | 2 | Skin traces mix |
| Skin low concentration (B) | 2 | Skin traces mix (Diluted 1-fold) |
| Feces high concentration (A) | 2 | Feces traces mix |
| Feces low concentration (B) | 2 | Feces traces mix (Diluted 1-fold) |
| Mixed 1 | 2 | Nasopharyngeal, Saliva, Skin, Feces |
| Mixed 2 | 2 | Nasopharyngeal, Saliva, Feces |
| Mixed 3 | 2 | Nasopharyngeal, Saliva |
| Mixed 4 | 2 | Saliva, Skin |
| Negatives | 6 | Buffer solution |
| Environmental samples Type Skin | 6 | Surfaces from: handlebars of bicycles or exercise bikes, gymnastic rings, keyboard |
| Environmental samples Type Saliva | 6 | Surfaces from: microphone, headphones, table cutlery |
| Environmental samples Type Nose | 6 | Surfaces from: used napkins, headphones, phone screen |

*2.2. Manual DNA Extraction*

An aliquot of each spiked sample (about 300 μL) was inserted into the semipermeable NAO Baskets and 20 μL proteinase K and 400 μL buffer AL were added. Each sample was slightly vortexed, and centrifuged at 10,000× *g* for 1 min, allowing the elution of the digestion solution. After incubation at 56 °C for 10 min and the addition of 400 μL ethanol, the washing step and DNA purification were performed in accordance with the manufacturer's instructions (QIAmp DNA Mini Kit and DNeasy Blood & Tissue Kit, Qiagen, Hilden, Germany). Finally, DNA elution was completed in 60 μL elution solution [10 mM tris(hydroxymethyl)aminomethane-hydrochloride and 0.5 mM ethylenediaminetetraacetic acid at pH 9.0], as previously described [42,50]. For pharyngeal biofluids and fomites samples, each swab was inserted into the semipermeable NAO Baskets and broken inside at the breakpoint. Approximately 200 μL lysozyme solution (20 mg/mL lysozyme, 20 mM tris[hydroxymethyl]aminomethane-hydrochloride at pH 8, 2 mM ethylenediaminetetraacetic acid, and 1.2% Triton X-100; Sigma-Aldrich, St. Louis, MO, USA) were added into the NAO Baskets and incubated at 37 °C for 30 min. Then, 20 μL proteinase K and 400 μL buffer AL were added, and the sample was centrifuged at 10,000× *g* for 1 min, allowing the elution

of the digestion solution. After incubation at 56 °C for 10 min and the addition of 400 μL ethanol, the washing step and DNA purification were performed in accordance with the manufacturer's instructions. DNA elution was completed in 60 μL elution solution [10 mM tris(hydroxymethyl)aminomethane-hydrochloride and 0.5 mM ethylenediaminetetraacetic acid at pH 9.0], as previously described [2,42,50]. The time for manual processing a set of 48 samples requires about 90 min, including aliquoting, incubation times, and centrifuging steps.

### 2.3. Automated DNA Extraction

A Nextrator-48S system (Genolution, Seoul, Republic of Korea) was used according to the manufacturer's recommendations after testing the already available kits for different purposes (e.g., viral RNA, yeast DNA, nucleic acids from stools and other matrices), in particular VN141R, MD141 or SD151 (Genolution, Seoul, Republic of Korea), optimizing the conditions in preliminary experiments. Briefly, preliminary tests included DNA extraction using the highest starting volumes (100–200 μL) and different lysis volumes (50–100 μL), finally suggesting loading 200 μL of the sample into an extract lysis volume of 50 μL (Supplementary Materials, Tables S1 and S2) in accordance with Best Practice Recommendations for Internal Validation of DNA Extraction Methods [54,55]. Experiments were finally performed using kit CVN291, showing the highest level of accuracy and repeatability for processing environmental swab samples diluted in collection and storage media (Table S1) [50,52–56]. Briefly, an aliquot of 200 μL was loaded into the well of the plate and extraction was performed using a program named VN in Nextractor-48S software system, following the manufacturer's instructions. At the end of the process, eluted DNA (50 μL) was collected and stored at −20 °C until the analysis. This automated extraction step can be performed on 48 samples in parallel within 21 min and within 45 min including the preliminary aliquoting and preparation steps.

### 2.4. DNA Yield and Purity

Following DNA extraction, the optical density of each sample was read at 260 nm and 280 nm using a Denovix spectrophotometer (Denovix, Wilmington, NC, USA). The total yield of extracted DNA was calculated by the DNA concentration (ng/μL) multiplied by the final elution volume (μL). Extraction efficiency was determined by the total DNA yield divided by the input volume (total DNA amount [ng]/input volume [μL]). Purity was compared based on A260/A280 absorbance ratios.

### 2.5. Analysis of mfDNA by Multiplex Real-Time PCR and Data Interpretation

Primers for different bacterial indicators and optimized reaction conditions were already established, as previously described [Table 2 and Table S2]. Briefly, amplifications were performed in four multiplex reactions to identify the different types of microflora present in the different samples: the 'mix skin' reaction was used to test for *Staphylococcus aureus* and *Staphylococcus epidermidis* present in skin microflora; the 'mix nasopharynx' reaction was used to identify *Propionibacterium* spp. and *Corynebacterium* spp. present in the nasopharynx; the 'mix saliva' reaction was used to test for *Streptococcus salivarius* and *Streptococcus mutans* present in the oral pharynx; and the 'mix feces' reaction was used to identify *Enterococcus* spp. *Bacteroides vulgatus* was present in fecal samples. Probes were labeled FAM/VIC/HEX with the BHQ-1 quencher. For each mix, samples were tested at least in triplicate. The amplifications were performed by programming the thermocycler (CFX96, Bio-Rad, Hercules, CA, USA) for 10 min at 95 °C and 40 cycles with 1 cycle consisting of 15 s at 95 °C and 1 min at 60 °C. For each sample, 11 μL template reaction was amplified. The PCR output was expressed as cycle threshold ($C_T$). Positive samples were those where ≥1 positive indicator ($C_T \leq 35$) was found in at least two mixes. Conversely, a microbial indicator was considered low-confidence positive with $C_T$ of 36 to 38 and negative when it was over the $C_T \geq 39$ threshold. Due to the use of recombinant and not native polymerase, for *Escherichia coli*, the criteria were modified as follows: Positive

samples were those where $\geq 1$ positive indicator ($C_T \leq 30$) was found in at least two mixes and a microbial indicator was considered low-confidence positive with $C_T$ of 31 to 35 and negative when it was over the $C_T \geq 36$ [2,42,50]. The $\Delta_{CT}$ is calculated using the automated extraction and manual protocol. The data for biological fluids were considered the average from experiments in duplicate, whereas the data from environmental samples represent the average of six samples with the same characteristics.

**Table 2.** Results of Interlaboratory validation of the real-time PCR approach.

| Type of Biological Fluids | Number of Samples | LAB 1 | LAB2 | $\Delta c_T$ Probe A | $\Delta c_T$ Probe B | Accuracy |
|---|---|---|---|---|---|---|
| Nasopharyngeal | 4 | 4/4 | 4/4 | 0.8 | 2.9 | 99.9% |
| Saliva | 4 | 4/4 | 2/4 | 3.4 | 3.3 | 75% |
| Skin sebum | 4 | 4/4 | 4/4 | 1.2 | 1.2 | 99.9% |
| Feces | 4 | 4/4 | 4/4 | 0.4 | 2.8 | 99.9% |
| Mixed 1 | 2 | 4/4 | 4/4 | 2 | 2 | 99.9% |
| Mixed 2 | 2 | 4/4 | 4/4 | 4.1 | 4.1 | 99.9% |
| Mixed 3 | 2 | 4/4 | 4/4 | 3 | 3 | 99.9% |
| Mixed 4 | 2 | 4/4 | 4/4 | 1 | 1 | 99.9% |
| Negatives | 6 | 6/6 | 6/6 | 0 | 0 | 100% |

### 2.6. 16S rDNA Amplicon Sequencing Analysis

Libraries for NGS were prepared according to the 16S Metagenomic Sequencing Library Preparation Guide (part 15044223 rev A; Illumina, San Diego, CA, USA). The PCR amplicons were obtained using Ba27F and Ba338R primers containing overhang adapters, as previously described in Table S3; [2,57]. Tagged PCR products were generated using primer pairs with unique barcodes through two-step PCR. According to the manufacturer's recommendations, the PCR-based amplification was performed using KAPA HiFi HotStart ReadyMix (Roche, Basel, Switzerland). Then, amplicons were pooled in equimolar concentrations of 50 pM. The libraries containing pooled-indexed samples with 10% spike-in PhiX (Illumina San Diego, CA, USA) control DNA were sequenced using iSeq 100 Reagent v2 (300-cycle).

### 2.7. Bioinformatic Analysis

The data generated from iSeq as raw reads in FASTQ formats were filtered using the Illumina 16S Metagenomics workflow and an in-house pipeline that was built on the Galaxy platform and incorporated various software tools to evaluate the quality of the raw sequence data (FASTA/Q Information tools, Mothur, ver. 1.46.0). Then, the high-quality sequences were clustered, and the operational taxonomic units (OTUs) with 99.9% identity were prepared based on using Ribosomal Database Project (RDP) and RDP's 16S Classifier 2.5 [58]. Observed OTUs were defined as observed species. A level of 97% sequence identity is often chosen as representative of a species and 95% for a genus.

### 2.8. Statistical Analysis

Relative abundances of community members were determined with rarefied data and summarized at each taxonomic level. The proportion of the microbiome at each taxonomic rank, such as phylum, order, class, family, and genus, was determined using the RDP classifier. Alpha and beta diversity were calculated using EstimateS software at a level of 97% sequence similarity (Estimates ver. 8.2.0). Regarding alpha diversity, the Shannon index and equitability index at the species level were computed [59,60]. The t-student tests were conducted to compare the yield of nucleic acids and the alpha diversity index between the samples subjected to different DNA extraction methods. An analysis of similarities (ANOSIM) analysis on beta diversity matrices was performed in QIIME to test for significant differences between the microbial communities according to the DNA extraction method. The significance of the ANOSIM test was assessed with 999 permutations.

## 3. Results

### 3.1. Protocol Aims and Main Steps

To analyze environmental samples for the presence of droplets or fomites at risk for indirect transmission of infections, a protocol was optimized to detect biological fluids through the identification of specific markers from their microbial signature (microbiota). The basic principle is founded on the identification of selected microorganisms by qPCR amplification of their genes [47,58]. The same qPCR procedure, know-how and equipment routinely used for COVID-19 clinical swab analysis was successfully transferred to fomite detection. Actually, in previous surveys, we applied it to environmental swabs and to simultaneous detection of viral RNA and fomites [2]. Recognition of droplets or biological fluids is a key indicator for indirect transmission risks through vehicles. Since the presence of droplets traces can suggest the presence not only of SARS-CoV-2 but also of other human respiratory pathogens, we focused on implementing a protocol for detecting fomites at risk on different surfaces. In a previous study we reported the possibility of detecting by qPCR both biological fluids and SARS-CoV-2 in environmental samples from hospital surfaces [2]. Here, we addressed critical issues related to scaling up the procedure, overcoming those difficulties related to the management of environmental samples rather than clinical samples (e.g., inhibitors, contaminants, etc.) [44,61]. The main key steps were summarized in (i) sample collection and storage; (ii) nucleic acid extraction; (iii) qPCR amplification; and (iv) interpretation of results, as reported in Figure 1.

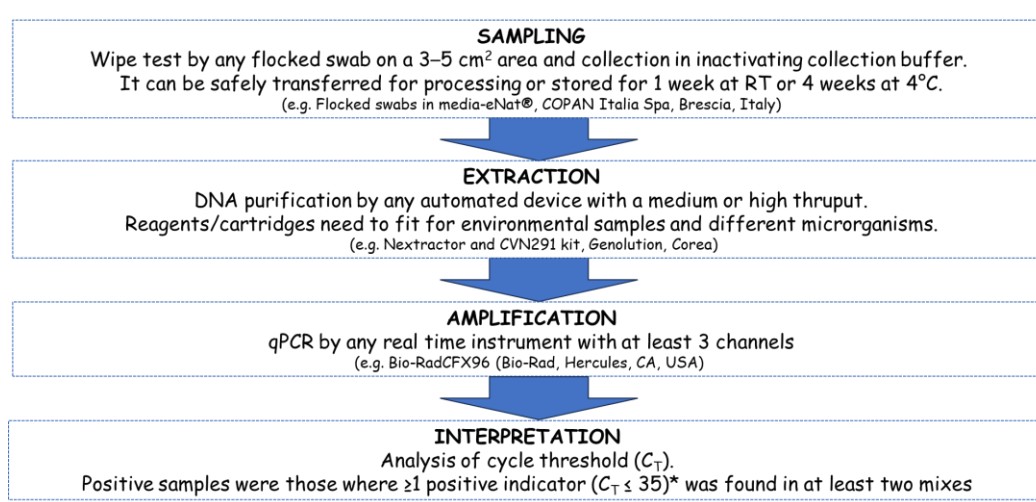

**Figure 1.** Schematic representation of the protocol for environmental monitoring of fomites by qPCR. The main phases include sampling, extraction, amplification, and interpretation of results. After selection of different options and intra- and inter-laboratory test, an optimized protocol was evaluated and proposed. The main steps and key issues are summarized. For environmental surveys, a platform with at least a medium throughput (>24 samples) should be considered to simultaneously process several samples to allow their epidemiological comparison in parallel. To maximize efficiency, the combination of automatic extractor and thermocycler should consider a similar number of samples to be processed in parallel. The specific equipment used in this study is also reported, considering at least three channels for the thermocycler in order to better manage all the different multiplex reactions simultaneously. * (for *E. coli* threshold is $C_T \leq 30$ due to the possible contamination of DNA traces in recombinant Taq).

Regarding step I, several solutions and materials are available for environmental specimen collection and have been considered for the wipe test (e.g., plastic, or microfibers tissues, cotton tips, waxes, parafilm, filter paper, nitrocellulose or nylon membranes, etc.). Finally, we focused on flocked swabs soaked in inactivating medium to allow safe transportation and appropriate storage (at room temperature for 1 week or in the refrigerator at 4 °C for 4 weeks, and up to 6 months at −20 °C to −80 °C), as previously reported [2,42,50].

Nucleic Acid Extraction (Step II) represented the bottleneck of the procedure. Manual processing is effective but not suitable for environmental surveillance due to the need of a fast and high-throughput processing approach to address multiple sampling points ($n$ = 5–10) from different areas (2 or more) of the same facility (e.g., hospital wards), because risk assessment and preventive actions are founded on the simultaneous comparison of the whole of the acquired data, usually tens or hundreds of samples [62]. Therefore, automatic devices at lower throughput (1–8 samples in parallel) are less appropriate with respect to medium-throughput devices able to process at least 40–100 samples simultaneously. Very high-throughput equipment (>700–1000 samples per day) and dedicated robots require additional maintenance and would be oversized at the present stage, unless considering a role for centralized laboratories in a future scenario. Even if any equipment for nucleic acid extraction can be considered, we concluded that a medium throughput instrument able to manage 48–96 samples within 1 h would fit the purpose if appropriate reagents for environmental samples are available or compliant. Steps III and IV are already very well established and can be performed with any kind of commercially available real-time PCR, fixing the correct fluorochrome for the specific filters, at least 3 channels for the FAM/VIC/HEX fluorochromes as carried out in the present protocol. Moreover, to maximize time and handwork, it is relevant to consider a thermocycle yielding the same number or multiples of the extraction device used in Step II (e.g., 96 wells amplification plate for 24–48 wells extraction cartridges). Interpretation of results is based on the comparison of Ct values and respect for the thresholds as usually performed for qPCR in SARS-CoV-2 tracing or as previously reported for detecting biological fluids using different markers [2,42,50].

### 3.2. Interlaboratory Validation of the qPCR Approach

Detection of fomites by qPCR is a reproducible and easily transferable method. Experiments were performed in parallel in different laboratories to assess the reliability of the qPCR approach for detecting biological fluids by applying the manual protocol on aliquots from the same samples using independent equipment and operators. Both laboratories received the kit and performed DNA extraction and amplification, showing high accuracy and reproducibility (Table 2). A total of 30 samples were analyzed in this interlaboratory test. Negative samples (collection media with unused swab) did not show any amplification signal at any concentration/condition, confirming the absence of background contaminations.

Both laboratories identified the right biological fluid with an accuracy of about 99.9%. However, some tests performed by an external laboratory (LAB2) on saliva samples partially failed to detect a positive signal for one of the markers (*S. salivarius*), giving a slightly lower level of accuracy of around 75% for that test. However, the total $C_T$ ($\Delta_{CT}$) difference between the two laboratories was not statistically significant ($p$ > 0.536).

### 3.3. Comparison Manual and Automated Protocol

3.3.1. Yield and Quality of Extracted Nucleic Acids

Once the qPCR approach using the manual protocol was validated, we successfully automatized the protocol. After intra-laboratory selection of appropriate reagents and optimization of experimental conditions for environmental samples (Table S1), a set of the same samples was processed in parallel by manual and automated protocols. The extraction efficiencies (yield) were 42 $\pm$ 16 ng/µL for the manual protocol, and 37.0 $\pm$ 19 ng/µL for the Nextractor system (Figure 1). The mean extraction efficiency of the automated approach was slightly lower than that of the manual protocol, although this difference was not statistically significant ($p$ = 0.156; Figure 2A). The A260/A280 absorbance ratios were approximately 1.9 $\pm$ 0.2 in all samples using both methods ($p$ = 0.900; Figure 2B).

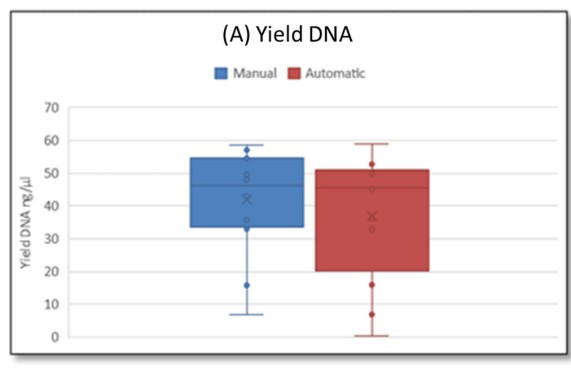

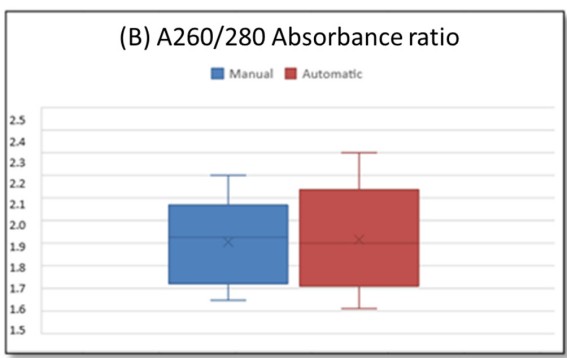

**Figure 2.** Comparison of (**A**) extraction efficiency (DNA amount/loading sample volume) and (**B**) A260/A280 absorbance ratio of extracted DNA among two methods.

### 3.3.2. Comparison by Real-Time PCR between the Two Extraction Methods

Automated extraction always provided amplifiable samples, even sometimes showing a higher sensitivity ($C_t$) than manual extraction, suggesting the detection of a slightly larger amount of DNA copies of the template or a better amplification efficiency ($\Delta_{CT}$). The correlation between the two extraction methods is over 99% ($p < 0.001$), indicating that the automated protocol is comparable to the manual one (as shown in Table 3). The same result was obtained when processing field samples obtained by environmental swabs from different surfaces.

**Table 3.** Comparison of qPCR results from manual and automated extraction protocols. * The $\Delta_{CT}$ is calculated between the automated extraction and manual protocol. ** The data for biological fluids represent the average from experiments in duplicate, whereas the data from environmental samples represent the average of six samples with the same characteristics.

| Type of Biological Fluids ** | Manual Protocol | Automated Protocol | $\Delta_{CT}$ * | Correlation |
|---|---|---|---|---|
| Saliva A—probe 1 | $31 \pm 0.2$ | $27 \pm 0.2$ | 4 | 0.99 |
| Saliva A—probe 2 | $27 \pm 0.2$ | $25 \pm 0.2$ | 2 | 0.99 |
| Saliva B—probe 1 | $33.2 \pm 0.2$ | $38 \pm 0.2$ | 5 | 0.99 |
| Saliva B—probe 2 | $28 \pm 0.1$ | $27 \pm 0.1$ | 1 | 0.99 |
| Nasopharyngeal A—probe 1 | $30 \pm 0.1$ | $30 \pm 0.5$ | 0.4 | 0.99 |
| Nasopharyngeal A—probe 2 | $34 \pm 0.2$ | $32 \pm 0.2$ | 2 | 0.99 |
| Nasopharyngeal B—probe 1 | $35 \pm 0.1$ | $32 \pm 0.5$ | 2 | 0.99 |
| Nasopharyngeal B—probe 2 | $33 \pm 0.5$ | $31 \pm 0.2$ | 2 | 0.99 |
| Skin Sebum A—probe 1 | $17 \pm 0.2$ | $16 \pm 0.8$ | 1 | 0.99 |
| Skin A—probe 2 | $17 \pm 0.6$ | $16 \pm 0.6$ | 1 | 0.99 |
| Skin B—probe 1 | $17 \pm 0.5$ | $18 \pm 0.5$ | 1 | 0.99 |
| Skin B—probe 2 | $18 \pm 0.4$ | $18 \pm 0.6$ | 0.2 | 0.99 |
| Feces A—probe 1 | $21 \pm 0.2$ | $19 \pm 0.2$ | 2 | 0.99 |
| Feces A—probe 2 | $21 \pm 0.2$ | $19 \pm 0.2$ | 2 | 0.99 |
| Feces B—probe 1 | $24 \pm 0.2$ | $26 \pm 0.2$ | 2 | 0.99 |
| Feces B—probe 2 | $24 \pm 0.4$ | $23 \pm 0.4$ | 1 | 0.99 |

**Table 3.** *Cont.*

| Type of Environmental Samples ** | Manual Protocol | Automated Protocol | $\Delta_{CT}$ * | Correlation |
|---|---|---|---|---|
| Environmental samples Type Saliva—probe 1 | 32 ± 0.1 | 30 ± 0.2 | 2 | 0.99 |
| Environmental samples Type Saliva—probe 2 | 30 ± 0.2 | 29 ± 0.3 | 1 | 0.99 |
| Environmental samples Type Nose -probe 1 | 33 ± 0.1 | 34 ± 0.2 | 1 | 0.99 |
| Environmental samples Type Nose—probe 2 | 30 ± 0.2 | 29 ± 0.3 | 1 | 0.99 |
| Environmental samples Type Skin—probe 1 | 32 ± 0.1 | 30 ± 0.2 | 1 | 0.99 |
| Environmental samples Type Skin—probe 2 | 30 ± 0.2 | 29 ± 0.3 | 1 | 0.99 |

### 3.3.3. Comparison of 16S rRNA Amplicon Sequencing between Two DNA Extraction Methods

The overall representation of the microbiota species in the extracted DNA is comparable between automatic or manual extraction protocols. Sequencing of the 16S rRNA gene was carried out on the 12 samples, in duplicate, using the Illumina iSeq100 platform, yielding a total of 1,437,709 reads with a mean read count of 119,809 per sample and a range of 17,513–157,776 reads (Figure 3, Table 4).

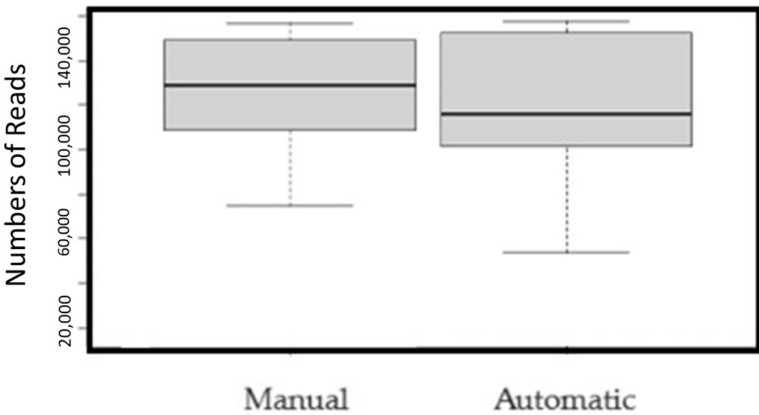

**Figure 3.** Comparison of the number of reads between samples extracted with two methods ($p = 0.001$).

**Table 4.** Summary of 16S amplicon sequencing analysis: raw data and phylogenetic diversity metrics.

| ID | Number Reads Pf Manual | Number Reads Pf Automatic | % Reads Pf Classified to Genus Manual | % Reads Pf Classified to Genus Automatic | Shannon (H) Manual | Shannon (H) Automatic | Otus Manual | Otus Automatic | Evenness Manual | Evenness Automatic |
|---|---|---|---|---|---|---|---|---|---|---|
| Saliva A | 17,513 | 18,954 | 98.12% | 98.00% | 0.798 | 0.751 | 136 | 138 | 0.28 | 0.30 |
| Saliva B | 148,172 | 152,661 | 99.56% | 99.60% | 0.771 | 0.753 | 442 | 450 | 0.27 | 0.27 |
| Nasopharyngeal A | 130,009 | 130,588 | 99.50% | 99.50% | 1.028 | 1.025 | 475 | 480 | 0.37 | 0.37 |
| Nasopharyngeal B | 99,065 | 99,068 | 99.41% | 99.35% | 2.242 | 2.242 | 547 | 554 | 0.82 | 0.82 |
| Skin A | 153,825 | 152,208 | 99.32% | 99.40% | 2.300 | 2.300 | 544 | 550 | 0.82 | 0.82 |
| Skin B | 152,201 | 155,529 | 99.19% | 99.20% | 2.414 | 2.398 | 672 | 680 | 0.82 | 0.81 |
| Feces A | 157,776 | 157,805 | 98.77% | 98.78% | 2.522 | 2.522 | 739 | 740 | 0.83 | 0.83 |
| Feces B | 121,322 | 121,343 | 99.41% | 99.45% | 1.697 | 1.698 | 491 | 500 | 0.65 | 0.65 |
| Mixed 1 | 75,481 | 53,501 | 98.84% | 98.83% | 2.055 | 1.703 | 456 | 382 | 0.77 | 0.63 |
| Mixed 2 | 129,933 | 103,949 | 90.96% | 91.06% | 2.625 | 2.555 | 658 | 515 | 0.79 | 0.80 |
| Mixed 3 | 119,846 | 110,912 | 99.58% | 99.57% | 0.077 | 0.099 | 252 | 294 | 0.02 | 0.02 |
| Mixed 4 | 132,566 | 107,778 | 99.02% | 99.01% | 2.454 | 2.251 | 617 | 666 | 0.82 | 0.72 |

Additionally, the alpha diversity indexes did not show significant differences in the number of observed species ($p = 0.2723$). The results were also consistent across different biodiversity measures, such as the OTU and Evenness indices, as reported in Table 4.

### 3.3.4. Beta-Diversity Analysis

To determine how different DNA extraction methods could affect the detection of microbial compositions in biological fluid samples, we conducted a PCoA analysis using Bray Curtis distances (Figures 4 and 5). The results showed that samples of the same type, extracted using the automatic or manual protocol, always clustered very close together (ANOSIM R = −0.076, $p < 0.001$). Therefore, both types of DNA extraction were consistent and allowed the correct assignment of each biological fluid trace (ANOSIM R = 0.557, $p < 0.001$).

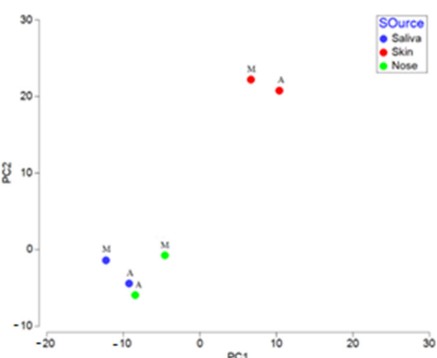

**Figure 4.** Principal coordinates analysis (PCoA) based on Bray–Curtis distances among the contaminated samples with two types of extraction. M: Manual protocol; A: Automated protocol.

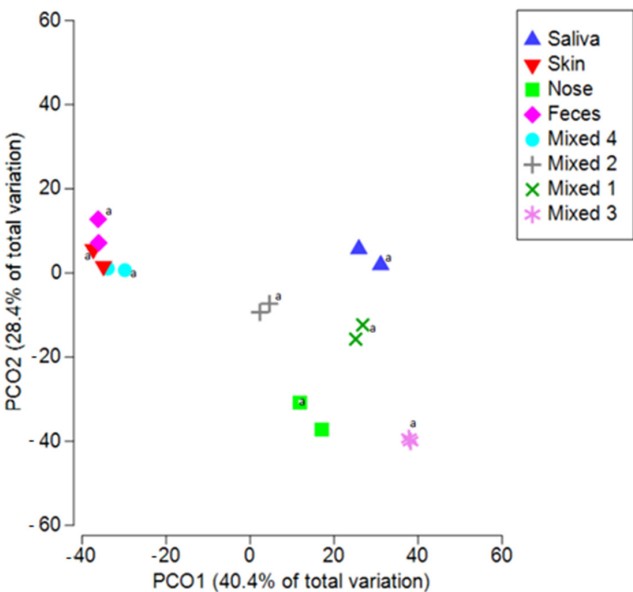

**Figure 5.** Automated and manual extraction provide overlapping results for different biological fluids. Principal coordinates analysis (PCoA) based on Bray–Curtis distances among all spiked samples (also mixed samples) with two types of extraction. Nose: Nasopharyngeal traces; Mixed 1: Saliva, Feces, Nasopharyngeal, Skin traces; Mixed 2: Saliva, Feces, Nasopharyngeal traces; Mixed 3: Saliva, Nasopharyngeal traces; Mixed 4: Saliva, Skin traces. (a: sample processed by automated extraction protocol).

The same results were obtained when testing samples with multiple contaminations. Indeed, when considering the Mixed samples (Figure 5), the differences between biological groups remained significant (ANOSIM R = 0.560, $p < 0.001$). Moreover, replicated samples are always clustered together, further supporting the reproducibility and reliability of the general protocol. Indeed, even if it is not necessary to evaluate all the microbiota biodiversity by NGS, it is enough to detect only a few markers by qPCR, these findings show that same results can be obtained when selecting different markers for addressing the same microbial signature of a given biological fluid.

Finally, the analysis of samples collected from the field from different surfaces confirmed the effectiveness of the approach, showing no inhibition in DNA amplification or major alterations in the biodiversity of the detected microflora. Therefore, we could exclude that different species (e.g., gram positives or gram negatives) could have influenced the extraction efficiency, ending up enriching or losing certain genomes during the extraction phase performed by the different protocols on environmental samples. Sequencing of the 16S rRNA gene on 18 environmental samples is reported in Figure 6, showing that differences between the biological groups are significant and consistent also when processing samples

from different surfaces and materials (ANOSIM R = 0.660, $p < 0.001$). Moreover, the difference between the extraction methods is not relevant (ANOSIM R = $-0.120$, $p < 0.05$).

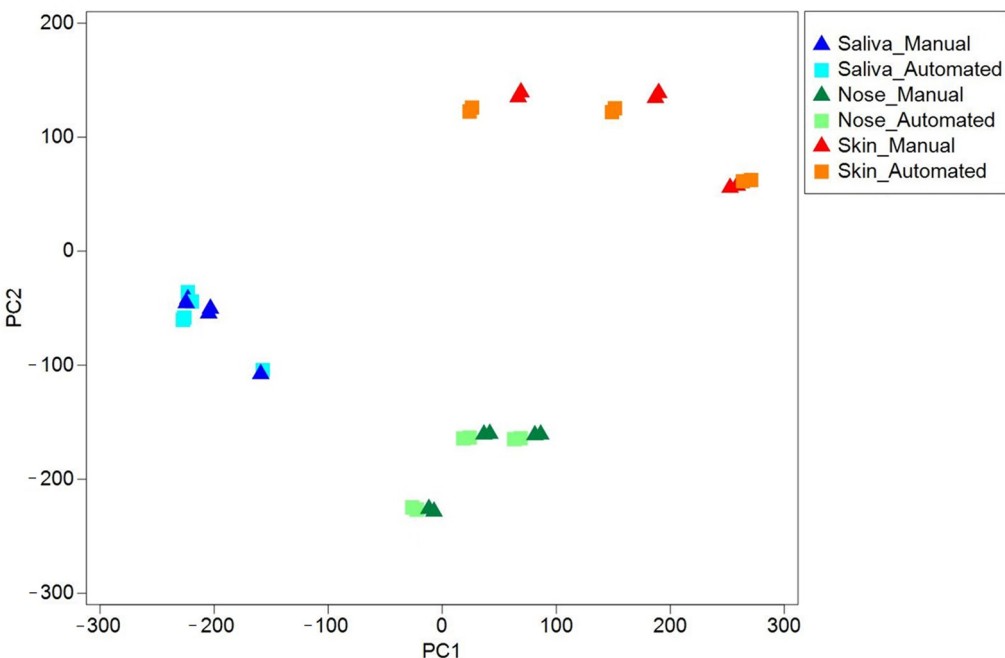

**Figure 6.** Biodiversity of field environmental samples extracted with automated protocol overlaps with manual protocol. Principal coordinates analysis (PCoA) based on Euclidean distances among all environmental samples with both extraction protocols. Saliva: Environmental samples type saliva (Surfaces with high saliva traces of contamination: microphone, headphones, table cutlery); Nose: Environmental samples type nose (Surfaces with high nasopharyngeal traces of contamination: used napkins, microphone, phone screen); Skin: Environmental samples type skin (Surfaces with high skin traces of contamination: handlebars of bicycles or exercise bikes, gymnastic rings, keyboard).

## 4. Discussion

Surveillance of environmental contamination represents a key issue in epidemiology, allowing addressing indirect transmission routes and monitoring the presence of fomites on surfaces at risk or verifying sanitation levels [1,63]. During the COVID-19 pandemic, significant efforts have been made to better understand the various pathways of SARS-CoV-2 transmission [1–6]. Studies have demonstrated that SARS-CoV-2 can be transmitted through droplets but have opened new questions regarding the role of different matrices, such as water or fomites, on different surfaces [1–14]. The availability of alternative and rapid methods for monitoring environmental contamination is strongly needed to face new pathogens or just to enforce hospital hygiene measures [61–63]. Environmental surveillance requires effective methods and the possibility to address a larger number of samples by simple high-throughput strategies. Here, we evaluated the reproducibility and reliability of fomite monitoring through the identification of residual organic debris or biological fluid traces by the detection of marker genes belonging to the human microbiota [2]. Critical steps include sampling from different surfaces, nucleic acid extraction, and amplification. An inter-laboratory test using the same sampling and qPCR approach showed a strong accuracy (99.9%) in detecting biological traces. The basic know-how and equipment were adapted from the routine clinical swab analysis test that was well-established and largely diffused during the COVID-19 pandemic to identify SARS-CoV-2 in nasopharyngeal samples. Therefore, the proposed protocol is promptly and highly transferable to different operators in different labs, exploiting the post-pandemic availability of equipment and know-how to perform qPCR molecular methods. We focused on transferring automatization for the SARS-CoV-2 nasopharyngeal swabs to further simplify the DNA sampling and purification steps for environmental samples. Droplets or other biological fluids potentially

able to carry pathogens were detected by characterizing the microbial signature by qPCR, as previously described [39,58]. This semi-automated protocol allows higher processability and faster monitoring of biological traces in the environment (Figure 7), making it possible to scale up, simplify and speed up the analysis of fomites in environmental swabs. This method enables the processing extraction of at least 48 samples within 45 min, which is twice as fast as the manual protocol handling 24 samples in at least 90 min, with less handling and time-consuming personnel involvement.

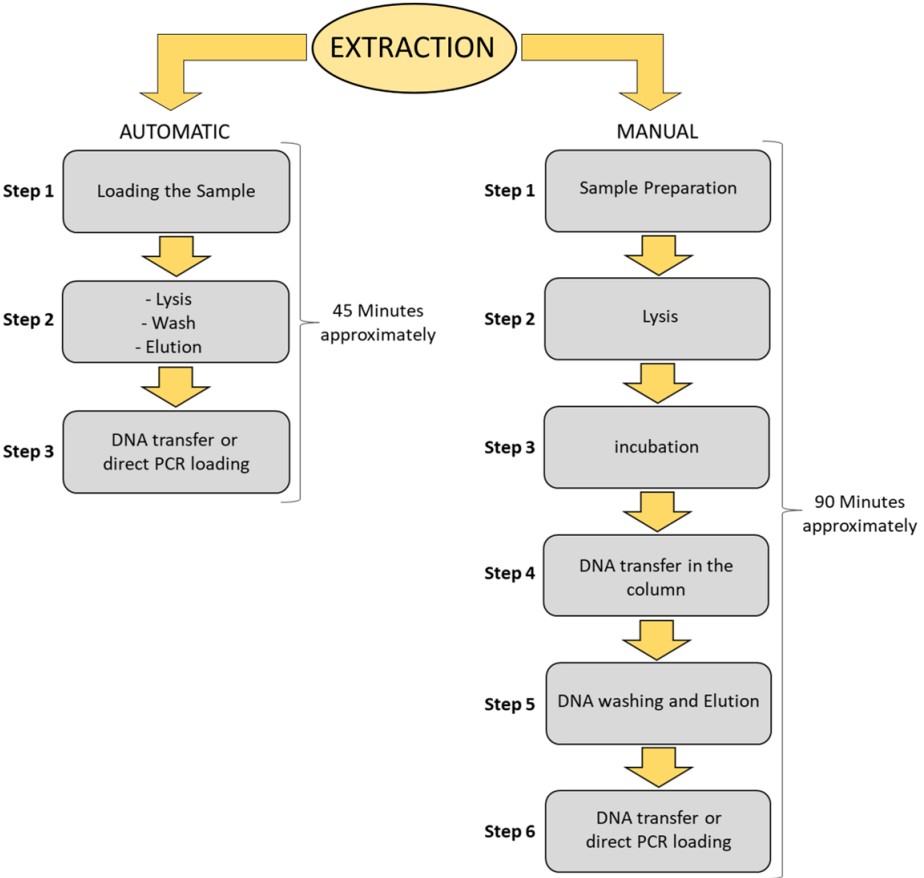

**Figure 7.** Automatic vs. manual extraction of environmental swabs. Flow chart with the main steps description of two extraction methods, steps, and times for 24 samples with manual protocol and 48 samples with automated protocol. Advantages include time, but also the lower involvement of personnel and the reduced risk of errors due to handling tubes. The same equipment used for COVID-19 tracing can be applied to environmental samples to detect droplets or fomites potentially contaminated with the pathogen. Addressing environmental samples can be challenging for a clinical laboratory because management of sampling and nucleic acid purification require a different approach. This molecular strategy is promptly available in most hospitals and laboratories that were already involved in contact tracing during the pandemic. This approach can overcome the bottleneck that has been slowing down epidemiological studies, hospital hygiene measures or surveillance actions based on access to simple and high-throughput environmental monitoring tools.

Advances in automatization provide a key advantage when processing many samples, reducing human errors in sample handling and devices and materials are more and more affordable, especially after the impulse provided by the COVID-19 pandemic on a worldwide scale. Indeed, coping with the pandemic provided a tremendous increment and diffusion of technologies dedicated to monitoring strategies by nucleic acid methods. To further assess the quality of NGS analysis of the extracted DNA and its feasibility in characterizing microbiota markers for detecting biological fluids or fomites, a comparison of microflora data was also performed by massive sequencing and bioinformatic

analysis. The whole of the results strongly supported the effectiveness of the extraction procedure, as it provides good quality DNA and comparable representation of species, as demonstrated by biodiversity indexes (Table 4, Figures 2–5). The use of automated DNA extraction by the Genolution Nextractor instrument, CVN291 cartridges and the optimized protocol was effective and even provided a slight increase in sensitivity ($p < 0.05$). However, the correlation between the manual and automatized extraction methods is over 99% ($p < 0.001$), so the automatic method appears to be comparable to the manual one, which is already well-established and diffusely applied. Therefore, the feasibility of approaching microbiota marker analysis was also confirmed by NGS analysis. There was no significant difference in the number of observed species between manual and automated methods ($p = 0.272$) and similar patterns were also observed in terms of the Shannon index and other biodiversity metrics. Also, when processing environmental samples (Figure 6), the difference between extraction strategies was not detectable ($R = -0.120$, $p < 0.05$). The whole of these observations supports the application of qPCR to detect biological fluids in fomites both after a traditional manual-based or an adapted semiautomated extraction. Surveillance and hospital hygiene do not require an elevated number of samples per day, as does the clinical laboratory. Therefore, a medium-size (24–48–96 samples/run) processing approach may represent a good compromise to allow fast and simple processing of tens or hundreds of samples in periodic hygiene monitoring or in focused surveillance actions. Our results may support environmental monitoring strategies by using already available protocols, knowledge, and equipment in a molecular biology Laboratory. However, several limitations need to be considered. Firstly, There are various systems available for nucleic acid extraction, but we have not yet optimized and transferred the protocol to all of these platforms. Moreover, the nucleic acid-based strategy allows identifying markers but cannot prove the vitality of the microorganism, because it also detects the genome from dead microorganisms. The method is cost-effective only in already equipped laboratories, and the study did not consider the economic impact of building a laboratory line for classical microbiology vs. molecular biology. However, in the last decades, know-how and equipment for qPCR have become widely diffused all over the world, also behind the push of the needs induced by the diagnosis or tracing of SARS-CoV-2.

## 5. Conclusions

The past few decades have been characterized by astonishing improvements in public health, but the COVID-19 pandemic provided a rapid leap forward, showing how new or reemerging microorganisms may threaten high-density, urban living as well as fragile populations. Thus, there is a vital need for identifying and monitoring the spread of new infectious diseases with new strategies focused not only on individuals but also on the environments, with special attention to hospitals, healthcare facilities or crowded facilities such as schools, barracks, public transportation, airports, or stations. Transferring methods from clinical routine to environmental epidemiology needs automatized methods and may represent an important and promising opportunity for prevention and public health. Here, we have proposed an improved strategy to surveil fomites in different environments.

**Supplementary Materials:** The following supporting information can be downloaded at: https://www.mdpi.com/article/10.3390/microbiolres15010008/s1, Table S1. Accuracy of DNA extraction in accordance with Best Practice Recommendations for Internal Validation of DNA Extraction Methods [52]; Table S2. Primers, probes and references [2,44,48,49,64].

**Author Contributions:** Conceptualization, V.R.S.; methodology, F.V., L.M.M., F.U. and G.G.; Formal analysis, F.V., L.M.M., F.U. and G.G.; data curation, F.V.; writing—original draft preparation, F.V.; writing—review and editing, F.V. and L.M.M.; supervision, V.R.S. All authors have read and agreed to the published version of the manuscript.

**Funding:** This study was partially funded by the MIUR-Fund-PON R&I 2014–2020 React-EU and IUSM Projects [CUP H83C23000160001; Prot. 1007-2023].

**Institutional Review Board Statement:** Not applicable.

**Informed Consent Statement:** Not applicable.

**Data Availability Statement:** Raw data will be made available if requested to corresponding author.

**Acknowledgments:** The authors thank GeneS Research Start Up (Rome, Italy) for providing equipment and reagents for sequencing and COPAN Italia Spa (Brescia, Italy) for providing FLOQSwabs® and eNat® collection medium for testing environmental sampling. A special appreciation to Genolution Inc. (Seoul, Republic of Korea) for providing technical information, equipment and for the qualified support to the study by preparing ad hoc cartridges to test the different preliminary conditions and optimize the automated protocol; We are grateful to Elena Scaramucci and Fabrizio Michetti for editing the manuscript and Manuela Camerino and Tiziana Zilli for library assistance.

**Conflicts of Interest:** The authors declare no conflicts of interest.

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
