# Peer review of "Automated Protocol for Monitoring Droplets and Fomites on Surfaces"

_2036-7481, doi:10.3390/microbiolres15010008_

Round 1

Reviewer 1 Report

Comments and Suggestions for Authors

The manuscript is interesting and explores a novel approach for analyzing microbiological entities on surfaces, thus could be useful for future researches. I have some issues to be confirmed by the authors:

1. In the abstract (L21-23), the authors stated that their present approach would allow the analysis of environmental swabs (48 samples) within 45 minutes (compared to 90 minutes for 24 samples using manual protocols). This statement should be justified within the discussion since based on the methodology described (L171-174), the analysis time would simply take 60 minutes.

2. If the findings were focused on SARS-CoV-2 viruses, then it would be better if the title includes such viruses.

3. L112: Please revise into Seoul, Republic of Korea

4. Figure S1: Since the figure is presented in the main manuscript, I believe there is no need to categorize the figure as a supplemented figure. 

Thank you.

Author Response

Reviewer 1

Comments and Suggestions for Authors

The manuscript is interesting and explores a novel approach for analyzing microbiological entities on surfaces, thus could be useful for future researches.

R: Thank you for your time in reviewing our work. Yes, that is the target of this manuscript, considering the results as an exemplificative application imposed at the time of COVID-19 pandemic, but easily extendable to the different situations, by a medium scale automatization. 

I have some issues to be confirmed by the authors:

  1. In the abstract (L21-23), the authors stated that their present approach would allow the analysis of environmental swabs (48 samples) within 45 minutes (compared to 90 minutes for 24 samples using manual protocols). This statement should be justified within the discussion since based on the methodology described (L171-174), the analysis time would simply take 60 minutes.

R: Yes, the whole time of the procedure is higher. However, this comparison of the required time was not related to the full process, but only to the “extraction step” and we excluded the following phases also because they are the same in terms of protocols and time. Therefore, it is not referred to L21-23 in chapter 2.5 but only to chapter 2.3. To avoid misunderstandings in the reader, we updated the text in the abstract, methods, and discussion. Thank you for your suggestion.

  1. If the findings were focused on SARS-CoV-2 viruses, then it would be better if the title includes such viruses.

R: Yes, the protocol was originally developed during the early COVID-19 pandemic for monitoring environmental surfaces for the presence of both fomites and SARS-CoV-2. However, nowadays, it represents a secondary issue, and it is only exemplary for other applications where both pathogen and its carrying biofluids could be simultaneously detected in the environment. We will stress this aspect in the discussion, adding some notes, but we would prefer not to add it in the title since in the general structure of this manuscript the data and methods were not explicitly reported. This would require extending the explanation to the simultaneous extraction of viral RNA and Bacterial DNA from the same swab, and this may confuse the reader. Therefore, we decided to just present the approach, focusing on the combination of these three innovative aspects: environmental detection of fomites, microflora/microbial signature approach, and automatization. 

  1. L112: Please revise into Seoul, Republic of Korea

R: Ok, done

  1. Figure S1: Since the figure is presented in the main manuscript, I believe there is no need to categorize the figure as a supplemented figure. 

R: Ok, thank you. We updated the text and the order of the figure numbers.

Thank you.

Reviewer 2 Report

Comments and Suggestions for Authors

The title "Automated protocol for monitoring droplets and objects" is non-informative and does not indicate the content of the work, which aims to assess the suitability and scaling of the implemented multiplex qPCR method for detecting droplets and objects in the environment.

The introduction emphasizes SARS-CoV-2, which is not the subject of this work. The introduction should concern environmental research and the interpretative value of the results obtained from such research.The work lacks coherence and systematization. The purpose of this particular environmental research is not explained in the article. The general statements presented do not add anything.

The results of the research are approximately doubling the scale of analyses, simplifying and accelerating the analysis of environmental swabs (processing at least 48 samples within 45 minutes compared to 90 minutes for approximately 24 samples using manual protocols).

Generally speaking, it is a test of the suitability (implementation) of existing biological material isolation protocols for environmental screening in order to simplify and increase the scale of environmental research. Commercial kits for isolation of biological material, which have not been described, were used to check the suitability. The criteria for selecting insulation putties and their differences are not described. Their dedicated matrices were not provided. There is no information on the number of trials used to verify suitability. Genetic identification (specific markers) of selected microorganisms that constitute part of the physiological flora was also performed. Therefore, it is difficult to conclude on this basis about the identification of a potential risk of infection, and the mere detection of the presence of physiological flora in the environment is only an indicator of cleanliness.

The proposed protocol is not shorter in time and cheaper than standard methods of culture and identification of microorganisms, with the advantage of identifying the spread of specific pathogens.

This is a methodological verification work with a very low degree of scientific novelty and is not described clearly. It requires many additions and explanations, as well as getting rid of over-interpretations. The purpose of the study and the method of achieving it are not precisely written. Many experiments have already been described in previous works to which the authors refer without providing key information in the article, such as the primers used or key information regarding the tests used. In many cases, the purpose of the experiments carried out and the limitations of the work that affect the results and conclusions obtained were not provided. No concise conclusions resulting from the work performed were distinguished or presented.

Comments on the Quality of English Language

Minor editing of English language required

Author Response

Reviewer 2

Comments and Suggestions for Authors

The title "Automated protocol for monitoring droplets and objects" is non-informative and does not indicate the content of the work, which aims to assess the suitability and scaling of the implemented multiplex qPCR method for detecting droplets and objects in the environment.

R: Dear referee, thank you for your time in reviewing this manuscript and for the suggestions. Even if the described approach can be used for direct monitoring of droplets and contamination on objects, actually, the title is limited to “Automated protocol for monitoring droplets and fomites on surfaces.”, approaching a wider question present in for hospital hygiene studies and related to the possibility to detect both pathogen and fomites. We could have compared it using different tools, but the specific novelty is related to the combination of all of these: innovative aspects: environmental detection of fomites by microflora/microbial signature approach extending the test through an automatization and testing its effectiveness in terms of correspondence to a gold standard reference, in this case one of the most diffused manual protocols for Nucleic Acid Extraction.

The introduction emphasizes SARS-CoV-2, which is not the subject of this work. The introduction should concern environmental research and the interpretative value of the results obtained from such research.

R: Yes, we agree and modified the introduction, accordingly. However, the study originated during the early COVID-19 pandemic, from the need to assess the presence of both SARS-CoV-2 virus and the carrying biological fluids (e.g. droplets) in environments at high risk (e.g. Piana et al 2021). We refer to that but did not show similar data neither the environmental monitoring flow chart and hygiene procedure, to avoid complicating further the description of the method, adding the co-extraction of viral RNA and bacterial (microbiota) DNA from the same sample. This approach worked very nicely, but it was not optimized and validated in such a way to be reported without creating confusion in the reader. Indeed, we should address the question of the co-extraction of DNA and RNA by using different protocols, the same cartridge, or different kits of reagents. Finally, the experience from SARS-CoV-2 remains a well-known and widely diffused background to support the need for a prompt strategy to monitor biological risks in the environment. We tried to better clarify this issue without going into the details. Thank you.

The work lacks coherence and systematization. The purpose of this particular environmental research is not explained in the article. The general statements presented do not add anything.

R: Fine, thank you. The purpose was described better simplified in the text at the end of the introduction.

The results of the research are approximately doubling the scale of analyses, simplifying and accelerating the analysis of environmental swabs (processing at least 48 samples within 45 minutes compared to 90 minutes for approximately 24 samples using manual protocols). Generally speaking, it is a test of the suitability (implementation) of existing biological material isolation protocols for environmental screening in order to simplify and increase the scale of environmental research. Commercial kits for isolation of biological material, which have not been described, were used to check the suitability. The criteria for selecting insulation putties and their differences are not described. Their dedicated matrices were not provided. 

R: We did not use insulation putties or classical swab matrices, but swabs based on flocked materials.  All details regarding the matrices were already reported in previous studies that we added to the text, which was improved accordingly. (This choice was an additional issue to make as suitable and acceptable as possible for environmental operators involved e.g. in hospital hygiene, based on already available tools, materials, and protocols).

There is no information on the number of trials used to verify suitability. 

R: Each test was performed at least in duplicate. To stress the method, the initial steps were performed in quadruplicate, (duplicate in parallel in two different labs by different operators). The study is still going on using the protocol successfully on hundreds of samples, providing very consistent, reproducible results. We did not go into details in the text, but the minimum information was provided (about at lines 185, 197, 358, 364).

Genetic identification (specific markers) of selected microorganisms that constitute part of the physiological flora was also performed. Therefore, it is difficult to conclude on this basis about the identification of a potential risk of infection, and the mere detection of the presence of physiological flora in the environment is only an indicator of cleanliness.

R: This is a crucial point. The general principle is the one traditionally used in hygiene: the approach based on “indicator”, therefore, a parameter that is not itself the hazard (e.g. the aetiologic agent for infectious disease), but a parameter that is related to the phenomenon under study (e.g. E.coli in water for assessing safety for drinking by determining the presence of possible enteric pathogens, or even better for our case: “an oral-fecal indirect transmission rout”). Therefore, the question of the genetic markers is just to detect the indicator (with specificity and sensibility, as previously we showed, e.g. in ref 47 and others). Cleanliness is part of hygiene, and it is definitely correct (and applicable to assess the effectiveness of sanification actions), but here, most of all it allows us to acquire information on the kind of biological fluid contamination, allowing us to consider the possibility to an indirect transmission route (e.g. by saliva, or nasal droplets…).

The proposed protocol is not shorter in time and cheaper than standard methods of culture and identification of microorganisms, with the advantage of identifying the spread of specific pathogens.

R: In our experience, cultural methods are much more expensive than molecular biology techniques, especially when a culture of virus is required. Definitely, we agree that molecular biology detection may provide (and provides, indeed) data from not vital and not infectious agents contaminating and environment, but, from an hygiene and public health point of view, the key question is related to its presence and the hypothesis that (now or in the past) it could have been infectious and however it arrived (and therefore it could arrive) on that surface. Nowadays, molecular methods are routinely considered and certified also for environmental microbiology analysis (e.g. Legionella in waters, on in food hygiene, e.g. ISO 67050), representing a marker/indicator of risk. In our case, the target and focus were on the risk of an indirect transmission. We hope to have provided some hints to better clarify the context we approached within the work reported in this manuscript. Finally, despite all of this, we agree your observation remains correct and the approach would not be a specific advantage when looking for the spreading (or infectivity/vitality) of an aetiologic agent in the environment, and that in this case a culture approach should be required.

This is a methodological verification work with a very low degree of scientific novelty and is not described clearly. It requires many additions and explanations, as well as getting rid of over-interpretations. The purpose of the study and the method of achieving it are not precisely written. Many experiments have already been described in previous works to which the authors refer without providing key information in the article, such as the primers used or key information regarding the tests used. In many cases, the purpose of the experiments carried out and the limitations of the work that affect the results and conclusions obtained were not provided. No concise conclusions resulting from the work performed were distinguished or presented.

R: Yes, we agree it is not a scientific work of basic science, but a very applicative and methodological presentation. Moreover, it is not aimed to a large community of researchers in microbiology, but it is focused on a very restricted group of scholars and operators involved in environmental monitoring within a public health frame. However, we understand that the manuscript has to reach the larger possible audience and most of all avoid misunderstandings in the reader, therefore we followed your precious indications and updated methods to better specify primer sequences, and some additional information on the used test, in a supplementary file. A section on the limitations of the study was added to the discussion. 

Thank you for your time and observations.

Round 2

Reviewer 2 Report

Comments and Suggestions for Authors

Thank you for substantive explanations of several comments. You have received some unclear feedback from someone who is not involved in environmental research, and there may be more such readers.

Information chaos for the reader was eliminated, especially by precisely defining the purpose of the study.

The limitations presented are great.

Final comments:

1. It would be good to provide the most important differences between the kits: VN141R, MD141, SD151 and CVN291, regardless of the reference to the source.

2. Line 169 - The table remained in the article despite being moved to the supplement.

3. "Each test was performed at least in duplicate. To stress the method, the initial steps were performed in quadruplicate, (duplicate in parallel in two different labs by different operators). The study is still going on using the protocol successfully on hundreds of samples, providing very consistent, reproducible results.” It would be worth adding this information in chapter "2.1. StudyDesign”.

4. It would be worth separating the conclusions from the discussion.

Author Response

65-Thank you for substantive explanations of several comments. You have received some unclear feedback from someone who is not involved in environmental research, and there may be more such readers.

Information chaos for the reader was eliminated, especially by precisely defining the purpose of the study.

The limitations presented are great.

R: Ok, thanks

Final comments:

  1. It would be good to provide the most important differences between the kits: VN141R, MD141, SD151 and CVN291, regardless of the reference to the source.

R: To avoid burdening the text, we have refrained from indicating any differences in the text and instead included the supplementary table (Table S2) with this information.

  1. Line 169 - The table remained in the article despite being moved to the supplement.

R: Ok, done

  1. "Each test was performed at least in duplicate.To stress the method, the initial steps were performed in quadruplicate, (duplicate in parallel in two different labs by different operators). The study is still going on using the protocol successfully on hundreds of samples, providing very consistent, reproducible results.” It would be worth adding this information in chapter "2.1. StudyDesign”.

R: Ok, thanks. Each test was performed at least in duplicate. To stress the method, the initial steps were performed in quadruplicate, (duplicate in parallel in two different labs by different operators). The study is still going on using the protocol successfully on hundreds of samples, providing very consistent, reproducible results. We did not go into details in the text, but the minimum information was provided (about at lines 185, 197, 358, 364). We have better underlined this in line 65-66.

  1. It would be worth separating the conclusions from the discussion.

R: Ok, done